# The specialized inner ear labyrinth of worm-lizards (Amphisbaenia: Squamata)

**Geneva E. Clark**[1], **Alessandro Palci**[2,3], **Rebecca J. Laver**[4,5], **Cristian Hernandez-Morales**[6], **Christian A. Perez-Martinez**[7], **Patrick J. Lewis**[1], **Monte L. Thies**[1], **Christopher J. Bell**[8], **Christy A. Hipsley**[9], **Johannes Müller**[10], **Ricardo Montero**[11], **Juan D. Daza**[1] *

**1** Department of Biological Sciences, Sam Houston State University, Huntsville, TX, United States of America, **2** School of Biological Sciences, University of Adelaide, Adelaide, SA, Australia, **3** South Australian Museum, Adelaide, SA, Australia, **4** Research School of Biology, Australian National University, Canberra, ACT, Australia, **5** University of the Sunshine Coast, Moreton Bay, Petrie, Queensland, Australia, **6** Department of Biology, The University of Texas at Arlington, Arlington, TX, United States of America, **7** Department of Biological Sciences, University of Missouri, Columbia, MO, United States of America, **8** Jackson School of Geosciences, The University of Texas at Austin, Austin, TX, United States of America, **9** Department of Biology, University of Copenhagen, Copenhagen, Denmark, **10** Museum für Naturkunde Berlin, Leibniz-Institut für Evolutions- Und Biodiversitätsforschung, Berlin, Germany, **11** Universidad Nacional de Tucumán, San Miguel de Tucumán, Tucumán, Argentina

\* juand.daza@gmail.com

**Data Availability Statement:** Amphisbaenian endocast 3D models are available from MorphoSource (https://www.morphosource.org, projectID: 000656777). Supportive Information S1

## Abstract

High-resolution computed tomography (HRCT) has become a widely used tool for studying the inner ear morphology of vertebrates. Amphisbaenians are one of the most specialized groups of fossorial reptiles but are poorly understood relative to other squamate reptile. In this paper we survey the anatomy of the inner and middle ear of these fossorial reptiles using HRCT models and we describe qualitatively and quantitatively (using 3D morphometrics) the anatomy of the inner ear. Amphisbaenians are diverse in skull anatomy, especially in the configuration of the snout, which correlates with digging modes. We demonstrate that the ear also exhibits a diversity of configurations, which are independent of phylogenetic relationships. Results from morphological analyses also allow us to describe 11 new potentially informative phylogenetic characters including some that help to diagnose amphisbaenians, such as: 1) the globular vestibule, ii) semicircular canals arranged in a circular trajectory, and iii) an extensive area of interaction between the columella footplate and the lagenar recess. Among extant amphisbaenians, *Rhineura floridana* has the most unusual inner ear configuration, including a horizontal semicircular canal that is in the same orientation as the inclined snout. The new morphological information helps us to better understand the morphology of headfirst-burrowing fossorial reptiles and contributes new data for resolution of phylogenetic relationships among amphisbaenians.

## Introduction

Members of the squamate clade Amphisbaenia (commonly known as worm lizards) are a group of highly specialized lizards with elongated bodies, rudimentary eyes, and, except for

and S2 includes the aligned coordinates used in the 3D morphometric analyses, and the list of species included in the analysis with their ecological variables. Additional details about the analysis are available at https://github.com/Nandezsendo/The-inner-ear-of-worm-lizards.

**Funding:** The Carl Gans Fund, Wilson-Warner, James D. Long and the Biology Department Scholarships supported Geneva E. Clark. Sam Houston State University Enhancement Research Grant (2014) to Juan D. Daza, Patrick J. Lewis, Monte L. Thies supported obtaining data for this study and funds from the Jackson School of Geosciences at The University of Texas at Austin to Christopher J. Bell. The funders had no role in study design, data collection and analysis, decision to publish, or preparation of the manuscript.

**Competing interests:** The authors have declared that no competing interests exist.

one genus (*Bipes*), they are completely limbless [1–3]. These reptiles adopted a subterranean mode of life and have developed one of the most curious locomotion modes among lizards–rectilinear locomotion–which is most often associated with snakes [4]. In relation to their subterranean habitats, amphisbaenians developed a diversity of adaptations that are reflected in their skeletal morphology [5], and soft tissue, including the loss of the eardrum [6].

Morphological descriptions of amphisbaenian skulls have mostly focused on cranial shape, particularly on modifications of the anterior portion [7–9], which is directly related to their subterranean lifestyle. The heads of amphisbaenians have diverse snout shapes, including rounded (e.g., most *Amphisbaena*, *Bipes*, *Blanus*, *Bronia*, *Cadea*, *Chirindia*, *Cynisca*, *Loveridgea*, *Trogonophis*, and *Zygaspis*), spade-headed (*Agamodon*, *Diplometopon*, *Pachycalamus*), keel-headed (*Ancylocranium*, *Anops*, *Baikia*, *Geocalamus*, *Mesobaena*), and shovel-headed (*Aulura*, *Dalophia*, *Monopeltis*, *Leposternom*, and *Rhineura*). The diversity in head shapes reflects variation in the primary surface contacting the substrate and the associated methods of burrowing [2, 3, 5], and documents convergent evolution across the different amphisbaenian clades [10, 11].

In contrast to the snout region, the posterior portion of the skull (i.e., the braincase and otic capsules) in fossorial groups remains poorly studied, especially the sensorial structures including the hearing and equilibrium organs. This produces a conceptual gap in understanding the evolution of the squamate ear apparatus [12], which functions in hearing and balance and is strongly linked to phylogeny [13]. One example is variation in the configuration of the columella (e.g., innervation, size, and orientation; Fig 1), whose structure and anatomical detail are largely understudied.

Digital endocasts of the inner ear labyrinth (inner ear for simplicity) are now a widely used source of data [14–16] for addressing a diversity of biological questions, including the fossorial or aquatic origin of snakes [13, 17], and the correlations among shape, ecology, and locomotion in limb-reduced skinks [18]. Convergent morphology is expressed in other burrowing and semiaquatic squamates [13] which presupposes that amphisbaenians should express a similar morphology [17]. Previous comparative analyses of the middle ear described amphisbaenians as having variation in the columella and presence of the extracolumella [19, 20], which further suggests that differences might also extend into the inner ear.

Among squamates, ground vibrations are transmitted to the inner ear (osseous labyrinth), which is formed by three major components: the semicircular canals, the vestibule (sacculus), and the lagena. In this paper we follow mostly Weber's terminology [21], but we prefer the term lagena instead of cochlea [or endosseous cochlear duct 15]—cochlea is a more adequate term for the lagena of mammals, in which this part of the inner ear is more spiral, analogous to a snail shell (*coclea* in Latin). The two terms are often used interchangeably and refer to homologous structures [22].

Amphisbaenians have a compact inner ear, that is the typical morphology of fossorial vertebrates [23, 24]; it is characterized by a large fenestra ovalis (which is consistent with the large stapedial footplate), a large spherical vestibule that occupies most of the space surrounded by the three narrow semicircular canals [6, 16, 21, 23, 25], and a lateral semicircular canal typically in contact with (or closely approaching) the sacculus [17]. A constriction usually demarcates a separation between the sacculus and the lagena [15, 26].

We examined morphological variation in the inner ear of amphisbaenians, with the intent to corroborate whether there is a consistent inner ear morphology, or alternatively, variation across clades. Ideally, variation would be studied in the context of ecological guilds, but given the cryptic nature of these reptiles, their microhabitat tolerances remain poorly documented, and knowledge about their burrowing adaptations are restricted to snout shape and digging methods [27] that are convergent across amphisbaenian species [3]. We drew from High-

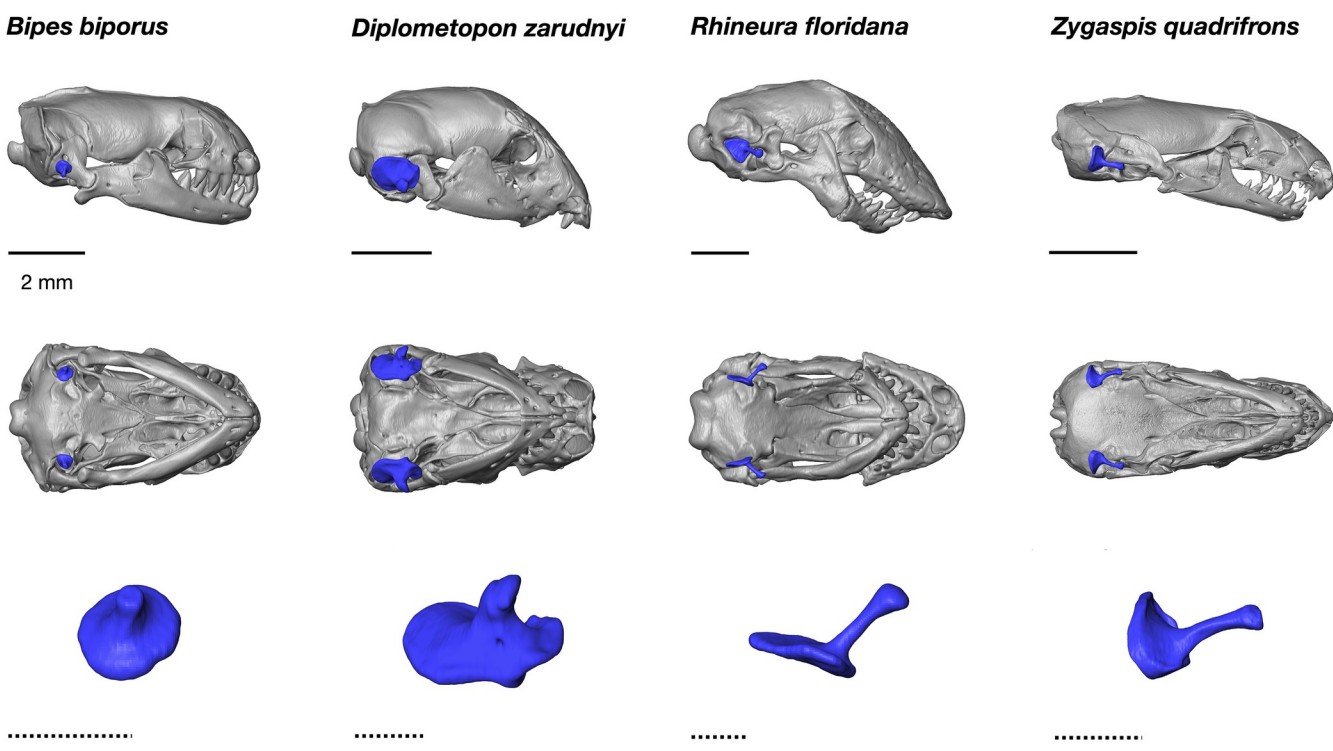

**Fig 1.** HRCT scans (top) showing variation in the morphology and position of the columella (rendered in blue) in the fenestra ovalis of four amphisbaenians, with skulls in left lateral and ventral views. Representative isolated columella in ventral view (bottom row), showing the orientation of the columella shaft: *Bipes biporus* (UF Herp 42060, ventrally), *Diplometopon zarudnyi* (UF Herp 68567, ventrolaterally), *Rhineura floridana* (UF Herp 180435, anterolaterally), and *Zygaspis quadrifrons* (FMNH:17751, anteriorly).

Resolution Computed Tomography (HRCT) data to generate endocasts of the labyrinth of the inner ear, and our approach included a combination of anatomical descriptions (qualitative comparison) and three-dimensional (3D) geometric morphometric (quantitative) comparisons to characterize morphological variation in the inner ear across amphisbaenians.

## Materials and methods

### 1. HRCT data collection

We included 19 amphisbaenian species, representing all the major clades [According to 3, 28]. Specimens were CT scanned at three institutions: The University of Texas High-Resolution X-ray Computed Tomography Facility (UTCT) located at Austin, Texas, USA; Adelaide Microscopy Facility at The University of Adelaide, at Adelaide, SA, Australia; and the Natural History Museum of Berlin, Germany.

Most specimens were scanned at UTCT in a Zeiss Xradia micro-CT. *Amphisbaena vermicularis* was scanned using a Skyscan 1076 micro-CT. *Blanus cinereus* and *Cadea blanoides* were scanned using a GE Phoenix Nanotom S. The species and specimens scanned are listed in Table 1.

### 2. Endocast rendering

Endocasts of the inner ears were rendered using the program Avizo Lite v9.5.0 (Thermo Fisher Scientific, 2018). The steps to create the endocast were as follows: 1) using the 'Interpolate'

**Table 1. Specimens used in geometric morphometric analyses.** CAS, California Academy of Sciences; FMNH, Field Museum of Natural History, Chicago; MCZ, Museum of Comparative Zoology, Harvard; MNMZB-UB, Natural History Museum of Zimbabwe, Bulawayo (formerly UM = Umtali Museum); TNHC, Texas Natural History Collections. MorphoSource and DigiMorph repository IDs are given, when available.

| Family | Species | Specimen ID | MorphoSource ID | DigiMorph ID | Collection Locality |
|---|---|---|---|---|---|
| Rhineuridae | *Rhineura floridana* | FMNH 31774 | 594371 | 31774 | Lakeland, Polk County, Florida, USA |
| Bipedidae | *Bipes biporus* | CAS 126478 | 594346 | 126478 | La Paz, Baja California Sur, Mexico |
| Bipedidae | *Bipes canaliculatus* | CAS 134753 | 594351 | 134753 | Rio Balsas, Guerrero, Mexico |
| Blanidae | *Blanus cinereus* | ZMB 29178 | 594356 | – | Valladolid, Castilla y León, Spain |
| Cadeidae | *Cadea blanoides* | ZMB 4082 | 594361 | – | Cuba |
| Trogonophidae | *Diplometopon zarudnyi* | FMNH 64429 | 594366 | 64429 | Qatif Oasis, Saudi Arabia |
| Trogonophidae | *Trogonohis wiegmanni elegans* | FMNH 109462 | – | 109462 | Morocco |
| Amphisbaenidae | *Amphisbaena alba* | FMNH 195924 | 594330 | 195924 | Serranía de Santiago Chiquitos Prov., Santa Cruz, Bolivia |
| Amphisbaenidae | *Amphisbaena caeca* | | – | – | Puerto Rico |
| Amphisbaenidae | *Amphisbaena fuliginosa* | FMNH 22847 | 594335 | 22847 | Canal Zone, Panama |
| Amphisbaenidae | *Amphisbaena vermicularis* | | – | – | |
| Amphisbaenidae | *Geocalamus acutus* | FMNH 262014 | 594341 | 262014 | Dodoma, Tanzania |
| Amphisbaenidae | *Zygaspis dolichomenta* | R-147 | – | 147B | |
| Amphisbaenidae | *Zygaspis dolichomenta* | R15907 | – | 15907B | |
| Amphisbaenidae | *Zygaspis ferox* | MCZ R-182217 | – | 182217A | Silverstreams, Chimanimani Distr., Zimbabwe |
| Amphisbaenidae | *Zygaspis kafuensis* | NMZB-UM 30040 | 594377 | 30040A | |
| Amphisbaenidae | *Zygaspis nigra* | FMNH 133021 | – | 133021B | Barotseland |
| Amphisbaenidae | *Zygaspis quadrifrons* | FMNH 17751 | – | 17751B | Botswana |
| Amphisbaenidae | *Zygaspis quadrifrons* | TNHC 85060 | – | 85060A | Koanaka Hills, Ngamiland, Botswana |
| Amphisbaenidae | *Zygaspis vandami arenicola* | FMNH 268569 | – | 268569A | Zimbabwe |
| Amphisbaenidae | *Zygaspis violacea* | FMNH 265728 | – | 265728A | Zululand, KwaZulu-Natal, South Africa |

function and a rectangle in the 'Selection' tool, we created a box around the inner ear; 2) with this box selected, we created an 'Inverse Selection' and added and locked this material; this creates a solid cast around the portion of the skull that bears the inner ear; 3) all the bone contained inside the cast was selected, added, and locked in a second material; and 4) with this new selection we used the 'Function' selection to fill all slices. Areas from the endocast that were missing (due to differences in density) were added manually to complete the model. We used a diversity of selection tools to make the endocasts, and in some cases we had to manually remove portions of bone that were added incorrectly to the model. Images of the inner ear were obtained in dorsal, lateral, and medial views, using the whole skull as a guide for orientation. Due to variable head morphology in amphisbaenians, establishing a standard plane of orientation is extremely difficult; we standardized the lateral orientation maintaining the basicranium horizontally as indicated in Fig 2. From that position, we rolled the head counterclockwise 90 degrees to get the dorsal view and finally 180 degrees to get the ventral view.

## 3. Landmarking & geometric morphometric analysis

We quantified inner ear shape variation by placing 47 landmarks (seven fixed and 40 sliding semi-landmarks, Fig 3) on surface files (.ply format) of the endocasts using the program Landmark Editor v3.6 [29], following a landmarking procedure previously developed for squamate reptiles [13]. To place amphisbaenians in the context of Squamata, we combined our new data with a previous squamate data set [13] for a total of 99 specimens, including 58 snake, 21 amphisbaenian, and 20 other lizard species. The sample also includes 22 generalist, 15 arboreal, 6 aquatic, 11 semiaquatic, and 45 fossorial species (S2 File).

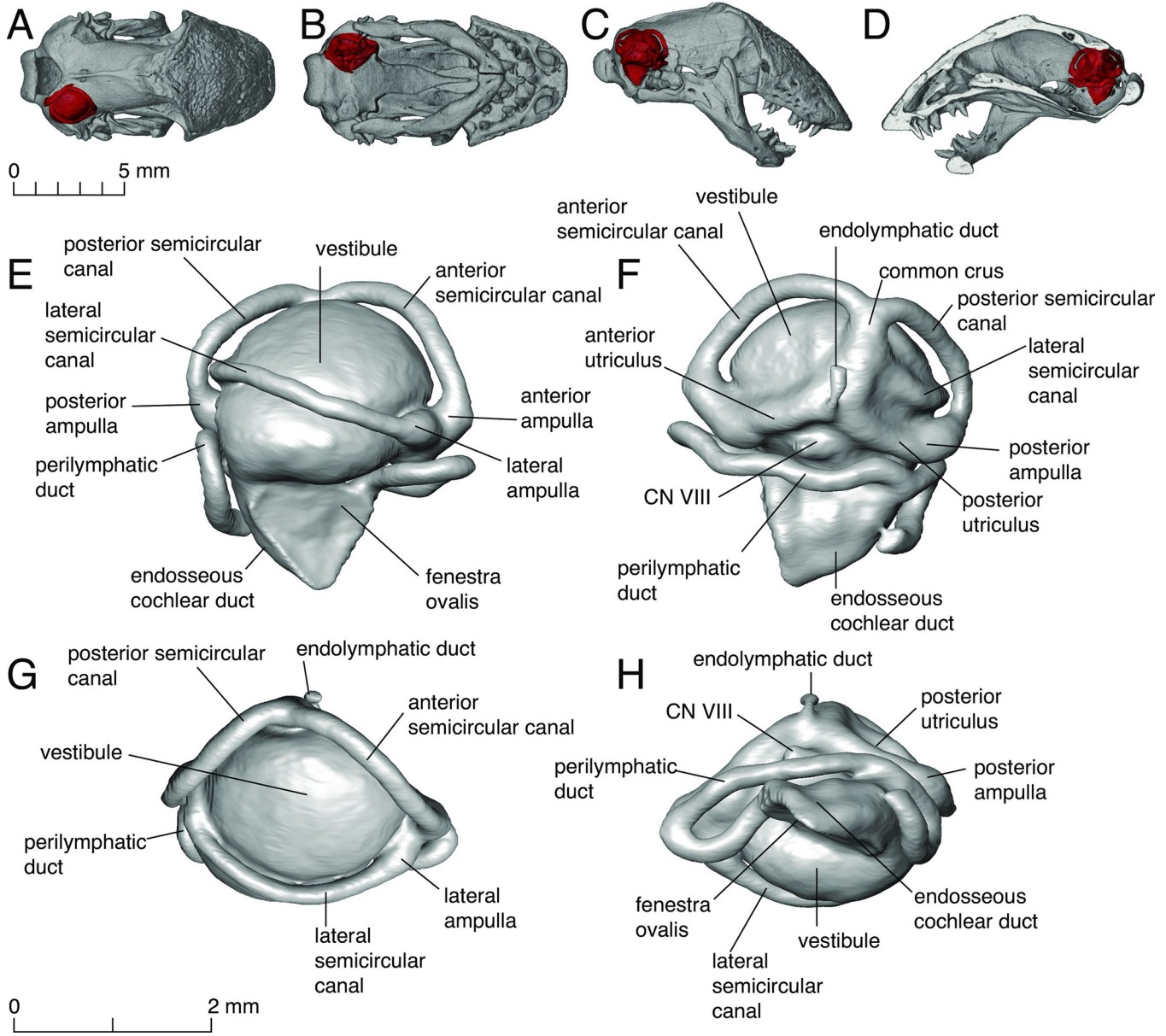

**Fig 2.** Skull of *Rhineura floridana* (FMNH 31774) showing the position of the right inner ear endocast in dorsal (A), ventral (B), lateral (C), and midsagittal (interior) (D) views. Detailed views of the endocast of the same specimen with parts labelled, in lateral (E), medial (F), dorsal (G), and ventral (H) views.

Landmark 48 (corresponding to the ventral end of the lagena) in the previous data set [13] was excluded, due to difficulty in establishing homology: in many amphisbaenians, the lagena is depressed with a large interaction with the columella footplate, and a fenestra ovalis that reaches the ventral margin of the lagenar recess. Also, note that landmark 47 was placed approximately in the center of the vestibule (Fig 3C); however, when this was not the case (i.e., the vestibule was not hemispherical), the landmark was placed in the circumcenter by placing it at a point that corresponded to the center of the circumscribed circle best fitting the outline of the vestibule.

Measurement error for the landmarking scheme was found to be negligible [13], hence it is not discussed further. The landmark configurations were scaled and aligned with a Procrustes

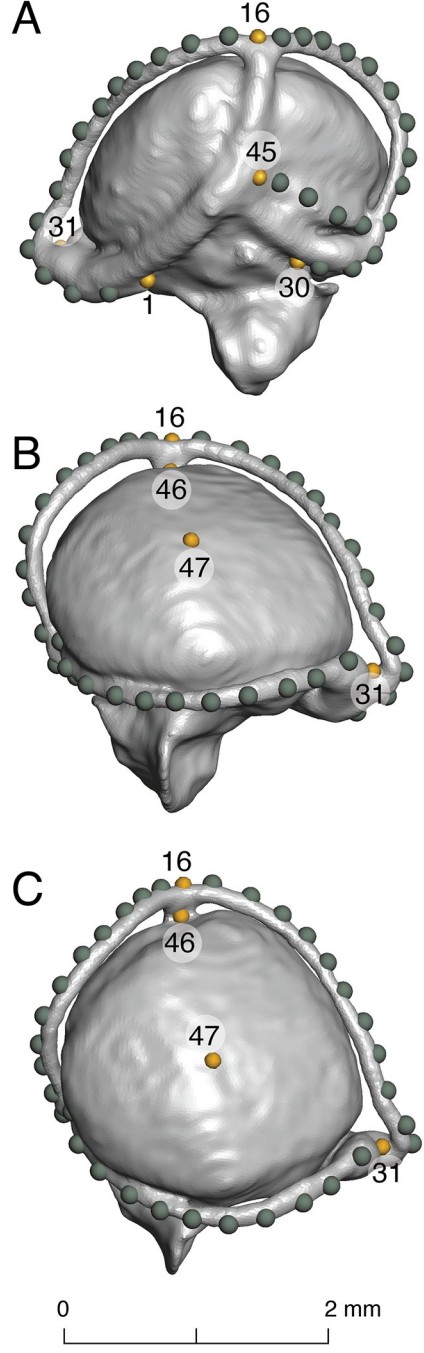

**Fig 3. Position of landmarks (yellow) and semilandmarks (green) on the inner ear endocast of *Cadea blanoides* (ZMB 4082).** Numbers indicate point landmarks on the medial (A), lateral (B), and dorsal (C) views.

superimposition using the package 'Geomorph' v3.0.5 [30] in R v3.3.3 (R Core Team, 2017; https://www.r-project.org/). Centroid size, defined as the square root of the sum of squared distances of each landmark to the centroid of the configuration, was used as an overall index of inner ear size. Principal Components Analysis (PCA) and statistical tests for correlation between centroid size and PC1 were performed using the same R package. Aligned coordinates

used for our analysis are available in the S1 File. Scripts and code used to run the analyses are available at https://github.com/cran/geomorph.

The endocasts of amphisbaenians were analyzed separately, including a subset that included phylogenetic information [31]. Phylogenetic signal was estimated using the parameter K, based on 1000 random permutations [32]. Finally, size related changes of the innear ear shape were tested using endocast size (centroid size) using phylogenetic generalized least squares (PGLS, type III) as implemented in geomorph (function procD.pgls).

## 4. Morphological comparisons and phylogenetic implications

Variation observed using qualitative and quantitative methods was expressed as unordered phylogenetic characters, and mapped using parsimony in the software WinClada, Version 1.61 (Asado) [33], characters were visualized as the transformation algorithms ACCTRAN and DELTRAN in an existing molecular scaffold [31], and including most of the amphisbaenians in this study (excluded the ones that have no molecular data). That phylogeny was estimated using five genes and extensive phenotypic data [34]. For our paper, we propose using these characters and explore their distribution, but they should be included in a more rigorously in a phylogenetic analysis to further test if they are useful in phylogenetic analyses. We also plotted the phylogeny using a subset of amphisbaenians, to better visualize the distribution of shape in the phylogeny.

## Results

Carl Gans and Ernest Glen Wever described the anatomy of the ear and hearing mechanisms in Amphisbaenia and other reptiles more than 40 years ago [5, 6, 21, 25]. Their model for the amphisbaenian ear can be interpreted as a three-component system: 1) a receptor, which in some amphisbaenians corresponds to the infralabial scales instead of the tympanic membrane; 2) a transmission segment, formed by two middle ear elements, the extracolumella and columella; and 3) a sound and equilibrium processing system, formed by structures located in the inner ear. Below we describe the middle and inner ear using two approaches: qualitatively, using a character-based approach and quantitatively, by studying the shape of the inner ear using 3D morphometrics.

## 1. Middle ear transmission segment

The extracolumella is not developed in the genus *Bipes*, but in others amphisbaenians is elongated (except in *Rhineura*). In species with an elongated extracolumella, it extends anteriorly from its posterior origin at the stapes and runs parallel to the jaw. In *Rhineura*, the extracolumella is oriented anteriorly as well, but instead of being long and slender, it is short and flattened (paddle-like) [35]. We explored the correlation of skull length and extracolumella in *Zygaspis*, one genus with an elongated extracolumella. We used the 3D renderings of left and right extracolumella in 15 specimens (skull length for smallest specimen – *Zygaspis quadrifrons*, 5.059 mm; largest specimen – *Zygaspis nigra*, 10.554 mm, skull length) [36]. The models were measured using Image J [37], the correlation between the logarithm of skull length and extracolumella length was isometric, with a slope near 1 for the entire sample. The slope was lower than 1 for the species in the sample with more specimens (n = 6), indicating negative allometry during the ontogeny: this size variation indicates that the extracolumella becomes proportionally shorter and stouter as the specimens increase in size (Fig 4). These changes in size are also reflected in shape, and the genus *Zygaspis* is known for having considerable morphological variation, including at least three types of extracolumella shapes, such as: spoon-shaped, ball-shaped, or linear [36, 38, 39].

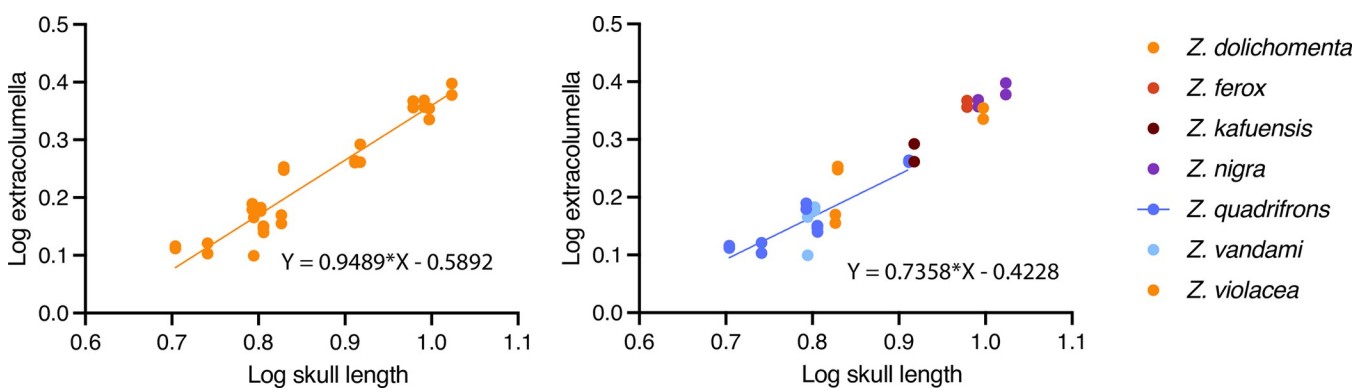

**Fig 4. Linear regression of the log of skull length and extracolumella.**

The columella fits into the fenestra ovalis in four different orientations (Fig 1). The columella in general has a short shaft and broad footplate; however, there is high variation in the shape of the columella, ranging from the shaft being directed laterally and contacting the extracolumella almost at a right angle (e.g., *Rhineura*), to a columella with a shaft abutting the extracolumella anteriorly (e.g., *Zygaspis*). Differences in the orientation of the columella span an almost continuous spectrum across species, but four broad categories can be identified: 1) columella footplate facing medially (*Rhineuridae*), 2) columella footplate facing dorsally (*Bipedidae*), 3) columella footplate facing ventromedially (Blanidae, Cadeidae, Trogonophidae, amphisbaenid *Geocalamus*), and 4) columella footplate facing posteriorly (amphisbaenids *Amphisbaena* and *Zygaspis*). These differences in columella orientation are correlated with the position of the fenestra ovalis in the braincase, which can be seen in the endocasts–Fig 5 shows the varying orientations of the fenestra ovalis in lateral view (using the standardized orientation, where the floor of the braincase is horizontal; also see Fig 1).

## 2. Inner ear sound and equilibrium processing system

Semicircular canals: Not all amphisbaenians exhibit the extreme compact shape of the osseous labyrinth previously described for fossorial vertebrates [17, 40]. The semicircular canals tend to be within close proximity of the vestibule in many species, and all amphisbaenian species develop the distinctive compact appearance reported in other groups such as uropeltid snakes [16], fossil snakes such as *Dinilysia patagonica*, and burrowing snakes such as *Xenopeltis unicolor* [17, 41]. In some species, there is a gap between the anterior and posterior semicircular canals and the vestibule (Figs 3 and 5). This spacing was observed in *Amphisbaena*, *Bipes*, *Cadea blanoides*, *Rhineura floridana*, and two species of *Zygaspis* (*Z. ferox* and *Z. nigra*). Another feature that characterizes other fossorial squamates is that the lateral semicircular canal is in contact with the sacculus [13, 16, 17, 26, 41]. This trait is present only in *R. floridana* and *Trogonophis wiegmanni* (Fig 5A and 5G), and to some extent in *Bipes* (Fig 5B and 5C), while in the other amphisbaenians, the lateral semicircular canal is shifted slightly ventrally and positioned closer to the constriction that marks the junction between the vestibule and the lagena (Fig 5). The lateral semicircular canal is anterodorsally oriented in *R. floridana* (Fig 5A) and to a lesser extent in Bipedidae (especially in *B. canaliculatus*, Fig 5C). In *R. floridana* the angle of orientation of the lateral semicircular canal matches the inclination of the snout. This same pattern of snout-lateral semicircular canal orientation is also present in the fossil *R. hatcherii* [35]; this remarkable correlation suggest ecological and biomechanical similarities between the extinct and extant species of *Rhineura*.

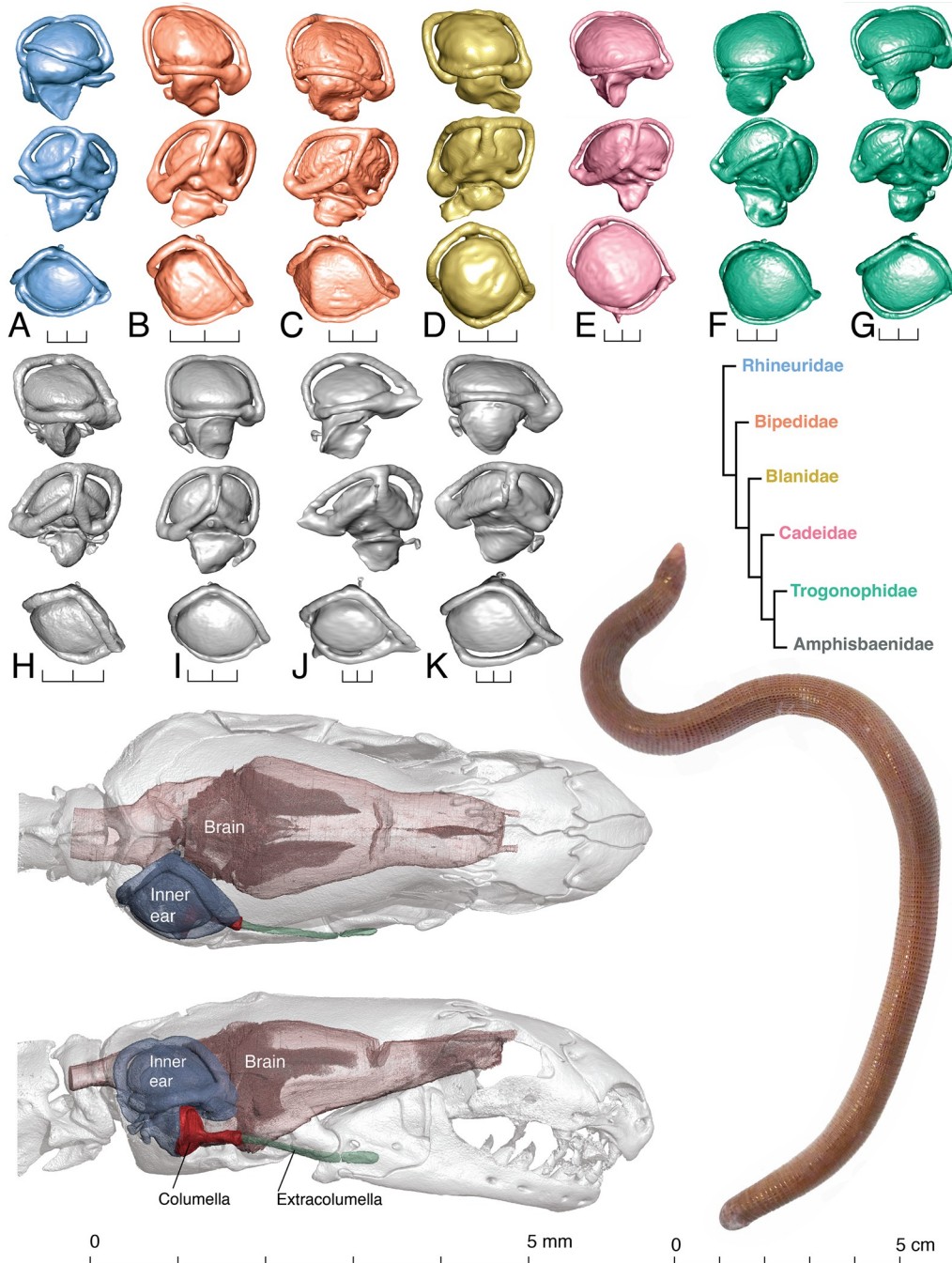

**Fig 5. Variation of inner ear labyrinths (right endocasts) in representatives from six families of amphisbaenians, with endocast color corresponding to family in the simplified phylogeny [31].** Each endocast is shown in three views (from top to bottom): lateral (anterior end to the right), medial (anterior end to the left) and dorsal (anterior end to the right): (A) *Rhineura floridana* (FMNH 31774), (B) *Bipes biporus* (CAS 126478), (C) *Bipes canaliculatus* (CAS 134753), (D) *Blanus cinereus* (ZMB 29178), (E) *Cadea blanoides* (ZMB 4082), (F) *Diplometopon zarudnyi* (FMNH 64429), (G) *Trogonophis wiegmanni elegans* (FMNH 109462), (H) *Zygaspis kafuensis* (NMZB-UM 30040), (I) *Geocalamus acutus* (FMNH 62014), (J) *Amphisbaena alba* (FMNH 195924), and (K) *Amphisbaena fuliginosa* (FMNH 22847). 3D models illustrated here are available on MorphoSource (Table 1). Scale bar for the endocasts equals 1 mm. Below, diceCT specimen of *Zygaspis kafuensis*, used to render the brain, inner ear endocast, columella and extracolumella. Live animal, *Amphisbaena schmidti* from Arecibo, Puerto Rico.

Another trait associated with other fossorial squamates is thinning of the semicircular canals. In general, among the amphisbaenians studied, this trait was most notable in *Cadea* and *Diplometopon* (Fig 5E and 5F). The semicircular canals in fossorial snakes are constricted, but in amphisbanians they are much wider openings [41], relative to body size, amphisbaenians have wider canals. In small amphisbaenians, the relatively large diameter of the semicircular canal can be attributed to a functional constraint upon these structures.

In *Rhineura*, the anterior ampulla and anterior semicircular canal are inclined in a sharp 90-degree angle (more evident in the medial view), while in other amphisbaenians, this junction is curved. In Cadeidae and Trogonophidae, the anterior ampullae are reduced. The anterior semicircular canal in *A. alba* is the most widely separated from the vestibule among the species we studied; it projects forward before curling back around to meet the anterior ampulla (Fig 5J). There is a similar pattern in *A. fuliginosa* (Fig 5K), although not as marked. In those species of *Zygaspis* examined, the anterior semicircular canal is at the same level as the anterior ampulla (Fig 5H).

Vestibule and lagena: We observed remarkable variation in inner ear structure across amphisbaenians, suggesting that size, mode of burrowing, and dwelling substrate might affect how these animals hear and process information about their position in space. In all specimens, the vestibule portion is proportionally larger than the lagena and in some forms, the size of the vestibule in lateral view is approximately eight times the size of the lagena (e.g., *Bipes canaliculatus*, *Diplometopon*, and *Trogonophis*), compared to about 4–5 times the size of the lagena in other species. Considering members of the genus *Zygaspis*, these proportions are affected by overall body size: in the largest form (*Zygaspis nigra*), the vestibule is about five times the size of the lagena, while in the smallest forms (*Zygaspis kafuensis*, *Z. vandami*), the vestibule is about three times the size of the lagena. However, the lagenar recess volume might be larger, as this structure seems to be depressed. In previous studies on other burrowing squamates [41], the vestibule is nearly spherical, and this is something observed in all species sampled, although with varying degrees of distortion. We also observed a well-marked ridge in the lateral surface of the vestibule of *Bipes* and *Blanus* (Fig 5B–5D).

Among amphisbaenians, the perilymphatic duct was found attached to the lagena in all species except *Rhineura floridana* (Fig 5A) and *Zygaspis kafuensis* (Fig 5H), in which it was a structure separate from the lagena. The posterior end of the perilymphatic duct in some forms develops a loop (e.g., *Zygaspis kafuensis* Fig 5H; and *Geocalamus acutus* Fig 5I).

*Rhineura floridana* differs from other amphisbaenians in the junction angle between the anterior and posterior utriculi, just above the acoustic recess. In that species, the angle is more obtuse, being nearly horizontal, while in the other species, the anterior and posterior utriculi join at an almost 90-degree angle (See medial views in Fig 5).

## 3. Geometric morphometric analysis

The first three principal components account for 64% of the variance in shape of the inner ear endocast (Fig 6). In the plot of PC1 vs PC2, the inner ears of amphisbaenians are clearly set apart from those of all other squamates, as they all have values for PC1 of less than -0.1. The only squamates that fall close to the amphisbaenian inner ear morphospace are the scincid *Acontias meleagris* (#2 in Fig 6), the anniellid *Anniella pulchra* (#10), the uropeltid snake *Teretrurus sanguineus* (#69), and the pipe-snake *Cylindrophis ruffus* (#25). Interestingly, all these forms share a fossorial ecology with amphisbaenians. In the plot of PC2 vs PC3, there are no clear distinctions for amphisbaenians, which fall around the center of the shape distribution, but with a narrow spread across PC3 and a much broader spread across PC2.

Negative values of PC1 primarily correspond to an almost circular configuration of the semicircular canals, especially on the lateral axis: this can be explained by a considerably

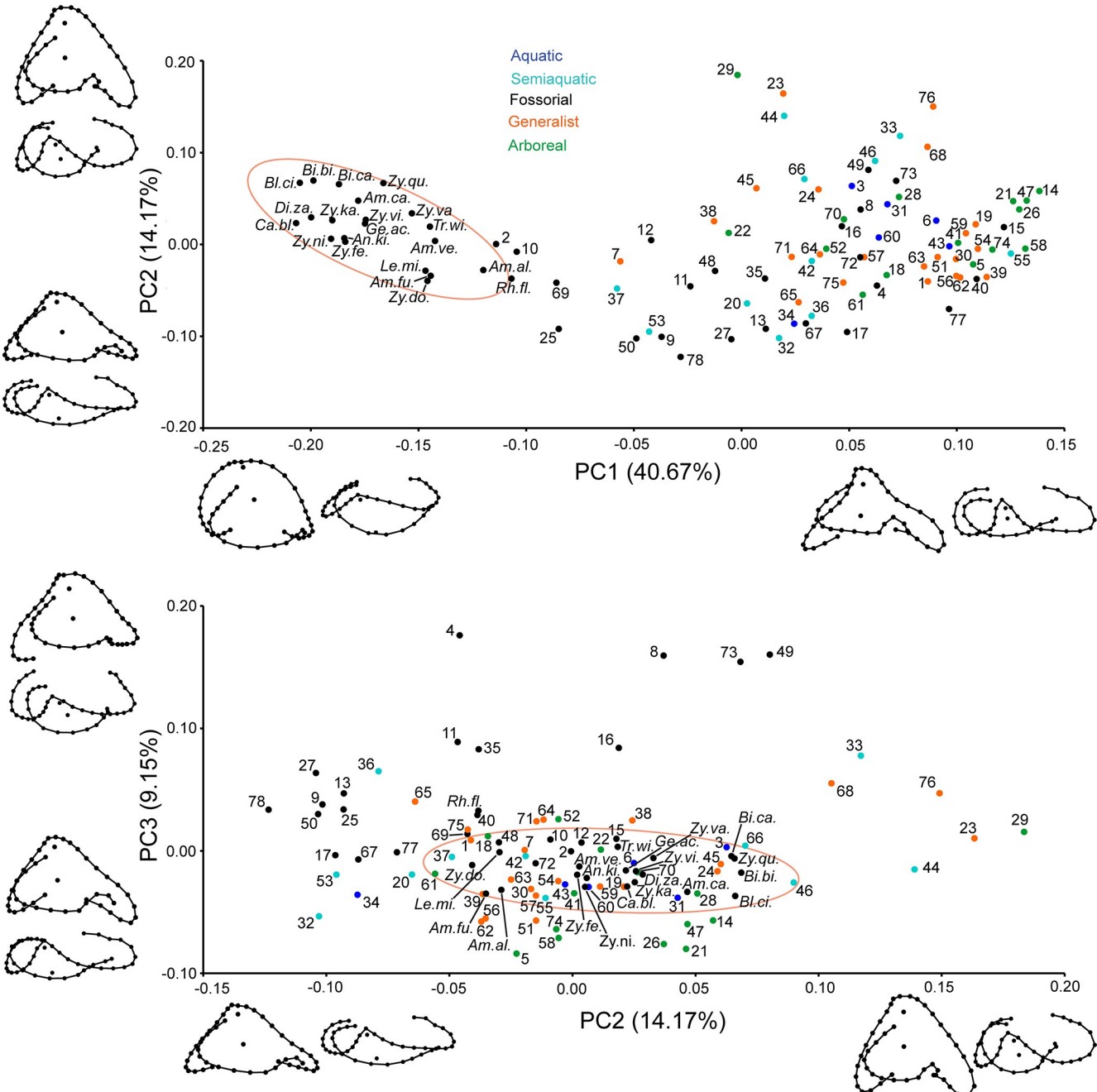

**Fig 6. Morphospace distribution of 99 inner ear endocasts of squamate reptiles defined by the first three principal components (ordinary PCA).** Projections of the Procrustes landmark configurations corresponding to the positive and negative extremes of each axis are also shown. The projections are in lateral (left or top) and dorsal (right or bottom) views with anterior to the right. The distribution of the amphisbaenians is highlighted by a 90% equal frequency ellipse (in pink). See S2 File for a complete list of species names that correspond to the numbers and abbreviations used in the plots.

inflated vestibule. Positive values, on the other hand, mark a pinching of the lateral profile of the anterior and posterior semicircular canals towards the baricentrum of the inner ear. This lateral profile is due to a change in orientation of the plane of each semicircular canal, which are both hemispherically arched and follow different courses. Positive values of PC1 are also associated with a relatively larger lateral ampulla, and a relatively smaller vestibule.

Negative values of PC2 correspond to changes from a globular (positive values) to a depressed (negative values) vestibule. Changes in the vestibule affect the position of the semicircular canals, for instance if the vestibule expands, the semicircular canals shift accordingly. Consequently, the anterior and posterior semicircular canals in forms with the highest values for PC2 appear shifted downward. Moreover, positive values for PC2 also mark a shift of the vestibule towards the posterior end, which is especially notable in dorsal view in Fig 4, and an anterior semicircular canal that is mediolaterally broader.

Negative values for PC3 capture the expansion or contraction of the lateral ampulla relative to the rest of the vestibule, and a posterior shift of the vestibule. Positive values of PC3 also mark a shorter anterior semicircular canal.

Finally, we tested for a correlation between inner ear size (approximated by centroid size) and shape variation (approximated by PC1) [42] and found that there is no significant correlation between the two, regardless of the statistical test used (Pearson: $t_{17}$ = -2.0792, cor = -0.4503, p-value = 0.05305; Kendall: T = 65, tau = -0.2398, p-value = 0.1637; Spearman: S = 1564, rho = -0.3719, p-value = 0.1176).

When endocasts of amphisbaenians were analyzed separately, to quantify the phylogenetic signal, the estimated parameter K was only 0.3993 (p-value = 0.023). This parameter indicates statistical independence of the inner ear shape from the phylogenetic relationships (Fig 7).

We did not find a significant allometric relationship between inner ear endocast shape and centroid size, using the PGLS comparative method (R2 = 0.11957 F(1,16) = 2.1729 P = 0.081), however, we found a significant allometric relationship when phylogeny is excluded (R2 = 0.31061 F(1,16) = 7.209 P < 0.001).

## 4. Potentially informative phylogenetic characters

Based on the observed variation and morphometric characters, 12 characters (11 of which are new) are proposed and consequently mapped onto the existing phylogeny of amphisbaenians (Fig 8).

1. Fenestra ovalis (lateral orientation maintaining the basicranium horizontally): (0) facing laterally (Fig 1, e.g. *Diplometopon*), (1) facing ventrally (Fig 1, e.g. *Bipes*), (2) facing anterolaterally (Fig 1, e.g. *Rhineura*), (3) facing anteriorly (Fig 1, e.g. *Zygaspis*).

2. Vestibule: (0) not globular, (1) globular (all amphisbaenians, Figs 2 and 5).

3. Vestibule: with lateral ridge: (0) absent (Fig 9A), (1) present (Fig 9B).

4. Semicircular canals: (0) not arranged in a circular trajectory, (1) arranged in a circular trajectory (all amphisbaenians, Fig 5).

5. Semicircular canals: (0) separated from the vestibule (Fig 9C), (1) close to the vestibule (Fig 8B).

6. Lateral semicircular canal: (0) at the vestibule-lagena junction (Fig 3B), (1) wrapping around the vestibule (Fig 2E).

7. Lateral semicircular canal: (0) mainly in the horizontal plane (Fig 3B), (1) anterodorsally oriented (Fig 2E).

8. Lagenar recess interaction with the columella footplate: (0) small portion representing the fenestra ovalis and columella footplate, (1) mostly to the size of the fenestra ovalis and columella footplate (all amphisbaenians, Fig 5).

9. Perilymphatic duct: (0) attached to the lagena (Fig 5B), (1) separated from the lagena (Fig 2F).

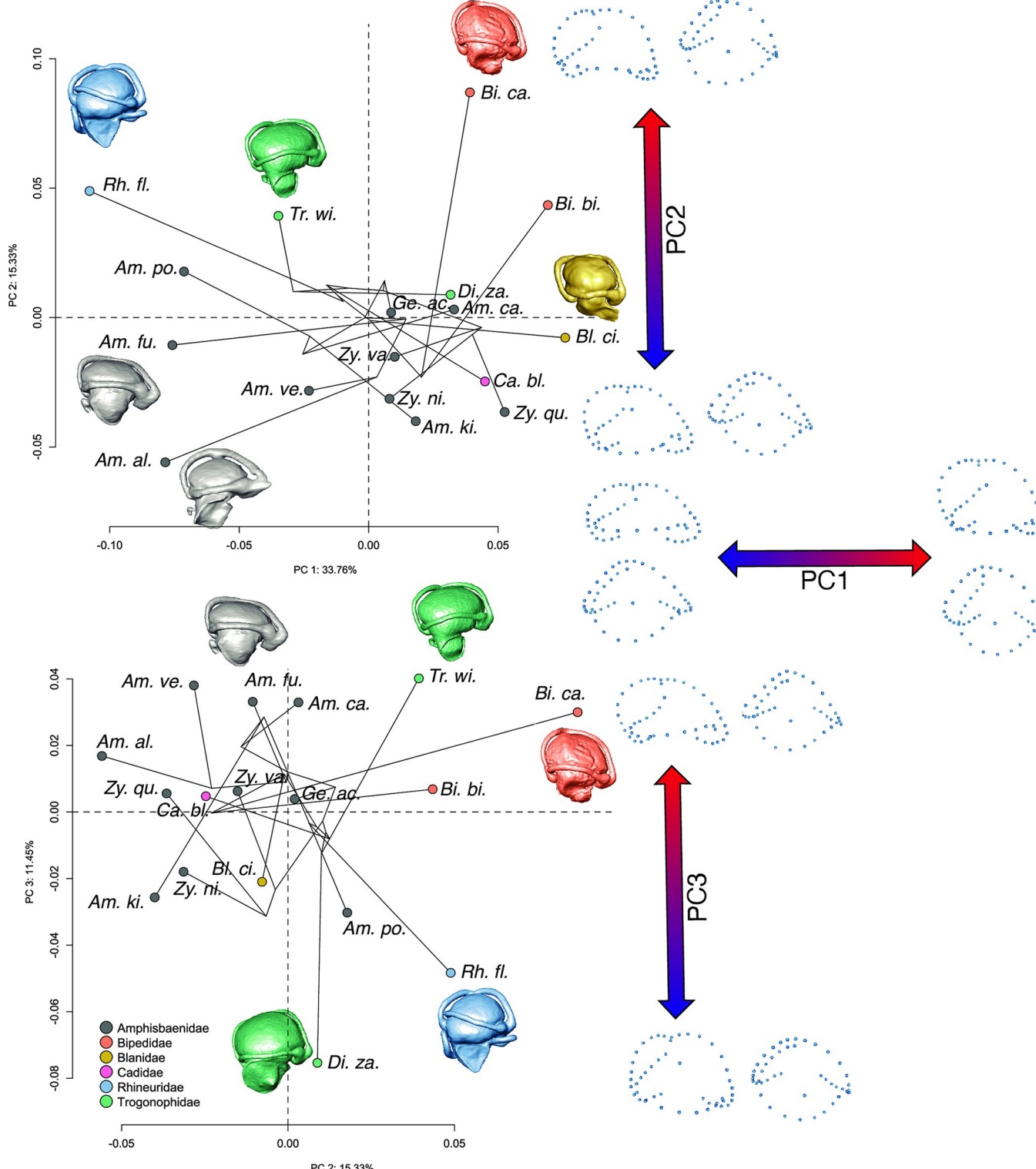

**Fig 7. Results of the Principal Components Analysis (PCA) of the 3D landmarks quantifying variation in inner ear shapes of amphisbaenians.**
Phylogenetic relationships plotted are subset from a published molecular phylogeny [31]. Top, PC2 plotted against PC1; bottom, PC3 plotted against PC2. Landmark transformations along the PC axes are shown to the right, where the landmark configurations are shown as if the right inner ear endocasts were observed in lateral (left or top) and dorsal (right or bottom) views. Data points in the PC plots are color-coded by family.

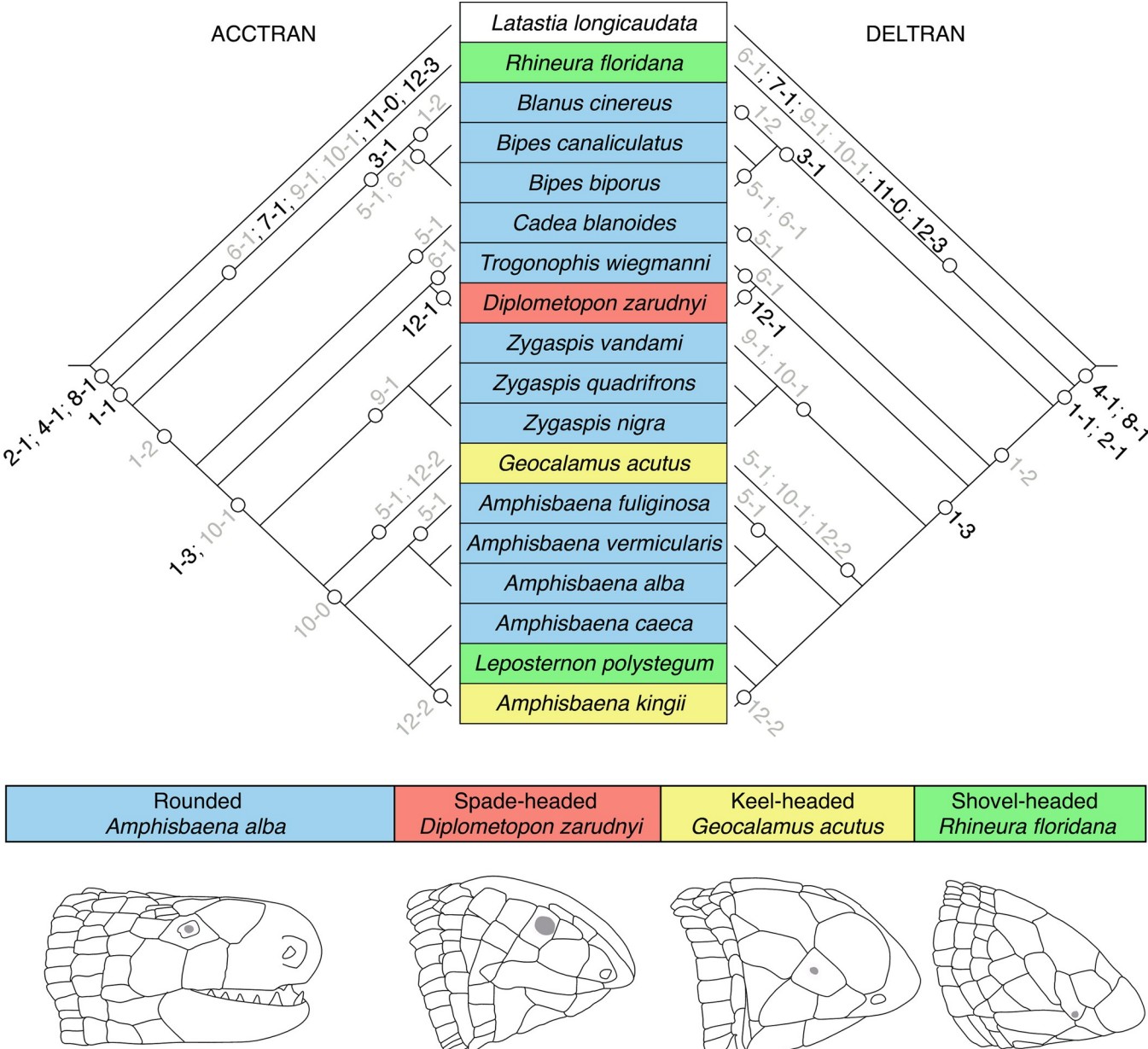

**Fig 8. Proposed characters mapped onto a pruned phylogeny of amphisbaenians [31].** Character descriptions are provided in the text.

10. Posterior end of perilymphatic duct: (0) without a loop (Fig 9B), (1) forming a loop (Fig 7A).

11. Anterior and posterior utriculus: (0) bordering the acoustic recess, dorsal border mostly horizontal (Fig 2F), (1) acoustic recess dorsal border deeply notched (Fig 5, all species except *Rhineura floridana*).

12. Additional character, head shape: (0) rounded, (1) spade-headed, (2) keel-headed, (3) shovel-headed (Fig 8). This character was described previously [5, 11]

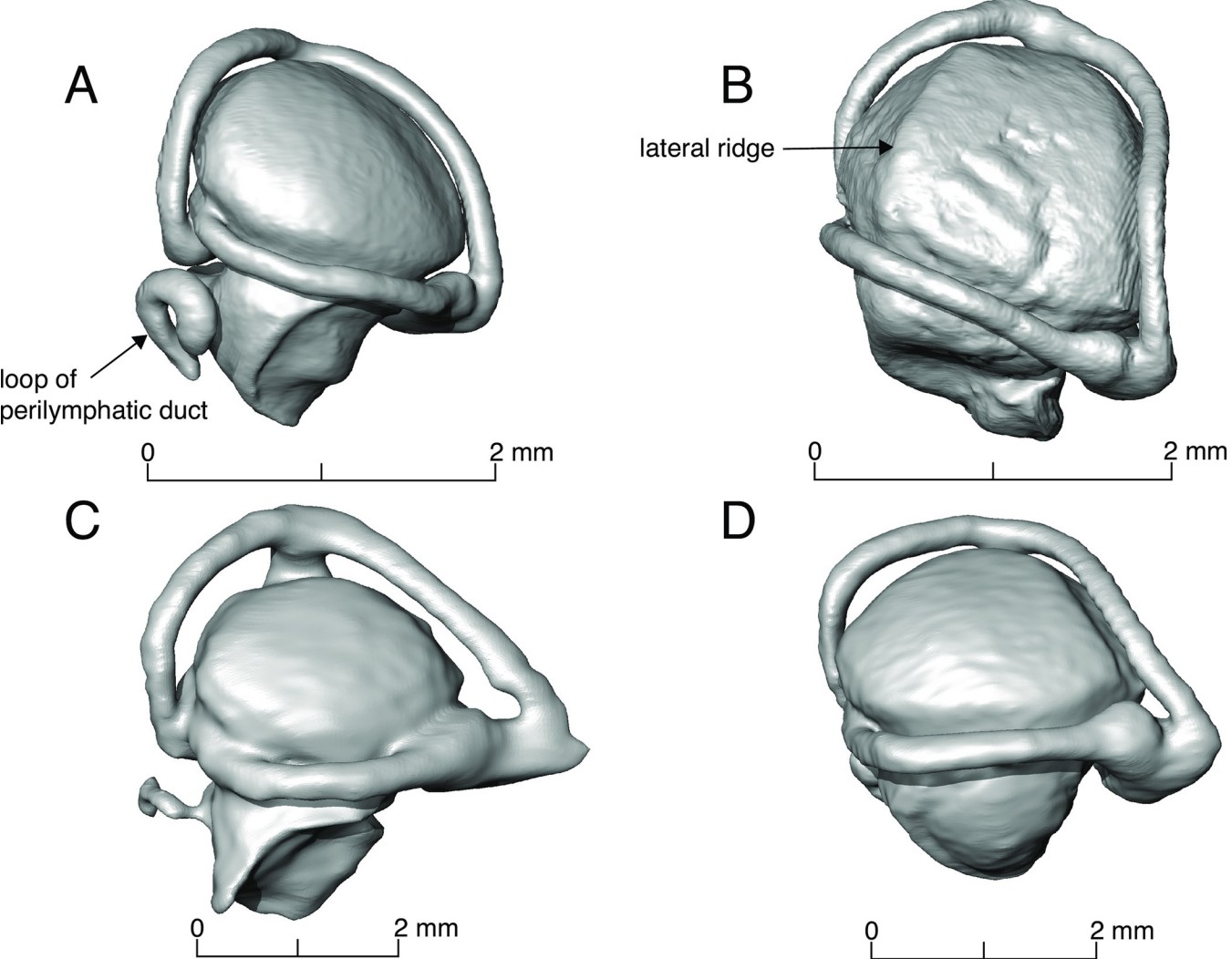

**Fig 9. Endocasts of the inner ear of some amphisbaenians illustrating character states proposed in this paper.** A) *Geocalamus acutus* (FMNH 262014), B) *Bipes canaliculatus* (CAS 134753), C) *Amphisbaena alba* (FMNH 195924), D) *Amphisbaena fuliginosa* (FMNH 22847).

## Discussion

Despite the fact that amphisbaenians are obligate burrowers and represent one of the most specialized fossorial group of squamates [13], their inner ear morphology is not uniform. Analysis of inner ear endocasts of a broad group of amphisbaenians illustrates considerable morphological diversity and offers a new source of potentially useful characters for phylogenetic analysis.

Previously, the compact shape of the inner ear was used as an argument supporting the fossorial ecology of the fossil snake *Dinilysia patagonica* [17], and considering its debated nested position within the crown group Ophidia [43, 44], it was used to support the argument for the burrowing origin of snakes [17]. This interpretation has been disputed, given that similar morphology is also present in semi-fossorial and semi-aquatic forms [13]. Previous works also demonstrated that many dedicated burrowing snakes have large spherical vestibules, although that does not seem to be the case in obligate burrowing blind snakes ("Scolecophidia") [41].

The presence of a large footplate in the columella and thin semicircular canals wrapped tightly to the vestibule are two other traits commonly associated with fossoriality.

Amphisbaenians have a large stapedial footplate. In fact, the fenestra ovalis area is the most prominent feature of the lagena. A globular and enlarged vestibule previously was proposed to be associated with fossoriality [17]. In this study, we found that amphisbaenians also have a globular and enlarged vestibule; however, they show marked differences, and this heterogeneity of inner ear morphologies could be correlated to the differences in the middle ear anatomy, including differences in the columella and extracolumella.

The amphisbaenians with the most expanded vestibule include Bipedidae, Blanidae, Cadeidae and *Diplometopon*. We previously stated that amphisbaenians inhabit localities with different soil types but, unfortunately not many studies have characterized in detail the type of substrate where each species dwells–additionally, it is known that some species choose specific soil types, while others can be found in multiple soil types [45, 46], making ecological associations to specific burrow properties difficult (e.g. *Rhineura*, which is found in sandy woodlands or sandhills [47]). A recent study taking into consideration the physical and chemical composition of soils found a correlation with head length and chemical soil composition (interspecific analysis), and variation in the width of prefrontal and frontal scales in association with clay and sand-soil contents (intraspecific analysis), the latter being attributed to digging force and friction [48]. Although the results of this study are encouraging, more studies, including detailed edaphic structure of the habitat of more amphisbaenian groups, are needed to establish correlation with skull, inner ear morphology, and digging patterns.

Given their cryptic nature, habitat preferences and soil tolerances of living amphisbaenians are difficult to describe accurately at this time, future studies should focus on investigating the behavior of amphisbaenians beyond descriptions of digging style. Additional data that would be invaluable include detailed information about the soil tolerance, microhabitat, depth of galleries, and biomechanics. Such data could facilitate drawing stronger correlations with morphological variation of the inner ear, because both the locomotion of amphisbaenians and transmission of ground-borne vibrations vary according to substrate type [49].

Amphisbaenians show striking convergence with limbless amphibian caecilians [40] – another group of actively head-first burrowing taxa that are also fossorial (except typhlonectids, which are aquatic caecilians) – in some of the middle and inner ear structures. In both groups, the stapedial footplate and fenestra ovalis are large. The vestibule is also globular and enlarged, which indicates parallel adaptations to perceiving ground-borne vibrations. One major difference between amphisbaenians and caecilians is that the semicircular canals in caecilians are thick and oriented in a different position, being mostly dorsal to the vestibule rather than surrounding it. However, the shared similarities of the inner ears of caecilians with amphisbaenians presupposes that their ears function in a similar way.

Our data provide new information about the diversity of inner ear morphology in one of the most specialized fossorial squamate groups and may help to improve our knowledge about the ecology of fossorial animals. At the same time our data suggest that comparisons among groups with similar ecology need to be done more carefully, given the unique combination of characters (morphological and morphometric) associated with fossoriality.

## Supporting information

**S1 File. Aligned coordinates used in the 3D morphometric analyses.**
(TXT)

**S2 File. List of species and habitat preference of specimens used in the 3D morphometric analyses.**
(XLSX)

## Acknowledgments

We thank Jessie Maisano and Matt Colbert from the University of Texas–High-Resolution X-Ray CT Facility for scanning most of the specimens used in this study. Alexandra Herrera from the Sam Houston State College of Osteopathic Medicine for her input on the allometry analysis. This project started as an independent undergraduate research project from Geneva E. Clark in 2017 at Sam Houston State University, and gradually grew into a larger survey.

## Author Contributions

**Conceptualization:** Alessandro Palci, Rebecca J. Laver, Patrick J. Lewis, Monte L. Thies, Christopher J. Bell, Ricardo Montero.

**Data curation:** Alessandro Palci, Cristian Hernandez-Morales, Patrick J. Lewis, Christy A. Hipsley, Johannes Müller.

**Formal analysis:** Geneva E. Clark, Alessandro Palci, Rebecca J. Laver, Cristian Hernandez-Morales, Monte L. Thies, Ricardo Montero, Juan D. Daza.

**Funding acquisition:** Geneva E. Clark, Alessandro Palci, Patrick J. Lewis, Christopher J. Bell, Christy A. Hipsley, Johannes Müller, Juan D. Daza.

**Investigation:** Geneva E. Clark, Christopher J. Bell, Christy A. Hipsley, Juan D. Daza.

**Methodology:** Geneva E. Clark, Juan D. Daza.

**Project administration:** Juan D. Daza.

**Supervision:** Juan D. Daza.

**Visualization:** Geneva E. Clark, Alessandro Palci, Cristian Hernandez-Morales, Christian A. Perez-Martinez, Juan D. Daza.

**Writing – original draft:** Geneva E. Clark, Alessandro Palci, Rebecca J. Laver, Cristian Hernandez-Morales, Christian A. Perez-Martinez, Patrick J. Lewis, Monte L. Thies, Christopher J. Bell, Christy A. Hipsley, Johannes Müller, Ricardo Montero, Juan D. Daza.

**Writing – review & editing:** Geneva E. Clark, Alessandro Palci, Rebecca J. Laver, Cristian Hernandez-Morales, Christian A. Perez-Martinez, Patrick J. Lewis, Monte L. Thies, Christopher J. Bell, Christy A. Hipsley, Johannes Müller, Ricardo Montero, Juan D. Daza.

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
