## [Decision Letter · Decision Letter 0]

28 Jun 2024

PONE-D-24-09623The unique inner ear labyrinth of worm-lizards (Amphisbaenia: Squamata)PLOS ONE

Dear Dr. Daza,

Thank you for submitting your manuscript to PLOS ONE. After careful consideration, we feel that it has merit but does not fully meet PLOS ONE’s publication criteria as it currently stands. Therefore, we invite you to submit a revised version of the manuscript that addresses the points raised during the review process.

 In addition to the reviewer's comments below, and in the sanititized copy attached, I'd like you to correct the following points:- Line 2: Short title: The inner ear. - Line 110: in a Zeiss Xradia micro-CT (too many capital letters). - Line 179: develop in the genus  - Line 329/330: The detected diversity should not only have utility in phylogenetic analysis, but should also be interpreted in a functional context. Please try to start such an interpretation in the light of different burrowing techniques of the studied genera. The discussion deserves an enlargement.- Line 336: workers have also demonstrated. - Line 342: of the of the - Line 347: with some species - Line 356: describe accurately - Line 373: diversity of - Line 375: "might help to improve our knowledge on fossorial ecologies": This sounds too general. Please try to give some hypotheses how the inner ear is connected to specific ecologies (see above). Convergencies with fossorial snakes should also better highlighted. References: Refs 16 and 25 are identical, also refs 13 and 17. Please change numbering. Figure Captions: Line 541: Rhineura floridana;   - Line 543: position of the right inner ear

We look forward to receiving your revised manuscript.

Kind regards,

Ulrich Joger

Academic Editor

PLOS ONE

Journal Requirements:

"The Carl Gans Fund, Wilson-Warner, James D. Long and the Biology Department Scholarships supported Geneva E. Clark. Sam Houston State University Enhancement Research Grant (2014) to Juan D. Daza, Patrick J. Lewis, Monte L. Thies supported obtaining data for this study and funds from the Jackson School of Geosciences at The University of Texas at Austin to Christopher J. Bell."

"We thank Jessie Maisano and Matt Colbert from the University of Texas – High Resolution X-Ray CT Facility for scanning most of the specimens used in this study. This  project started as an independent undergraduate research project from GEC in 2017 at Sam Houston State University, and gradually growth into a larger survey. Funding was received from multiple agencies, but we want to specially thank The Carl Gans Fund for sponsoring GEC to present part of this work in the 2016 meeting of the American Society of Ichthyology & Herpetology."

"The Carl Gans Fund, Wilson-Warner, James D. Long and the Biology Department Scholarships supported Geneva E. Clark. Sam Houston State University Enhancement Research Grant (2014) to Juan D. Daza, Patrick J. Lewis, Monte L. Thies supported obtaining data for this study and funds from the Jackson School of Geosciences at The University of Texas at Austin to Christopher J. Bell."

**Additional Editor Comments:**

The reviewer suggested a number of valuable improvements for your manuscript. Please try to follow them, especially the discussion should be enlarged as she proposed.

Reviewers' comments:

Reviewer's Responses to Questions

**Comments to the Author**

1. Is the manuscript technically sound, and do the data support the conclusions?

Reviewer #1: Yes

2. Has the statistical analysis been performed appropriately and rigorously? 

Reviewer #1: Yes

3. Have the authors made all data underlying the findings in their manuscript fully available?

Reviewer #1: No

4. Is the manuscript presented in an intelligible fashion and written in standard English?

Reviewer #1: Yes

5. Review Comments to the Author

Reviewer #1: This study represents a really neat example of using CT technology to access anatomical systems that are not usually observable, in taxa that also are not often easily observable! I also appreciate that qualitative comparisons were not overlooked, but utilized in tandem with popular 3D geometric morphometric methods.

Overall the paper was well-written with clear anatomical descriptions and analysis. The syntax and grammar can be improved in many places, and all wording suggestions, typo edits, requests for citations, etc. have been annotated directly onto the attached PDF, along with all comments and questions. There are also areas of text where some re-structuring or re-organization will be necessary to improve the flow and clarity of the paper. The full resolution CT and other imagery was sharp and clear, a real strength of the paper; the color coding is clever and helpful for following along in the figures.

The more major revisions are repeated and highlighted here for emphasis:

-abstract: suggest including more of the specific (and useful!) results observed in this study.

- line 67: innervation and shaft morphology not really discussed in the Results though mentioned here as needing study; can more info about the shafts and foramina be added to the paper? The columellas are nicely figured.

- The mapping of new characters onto the amphisbaenian tree is interesting and could be useful in future. Please add a little more detail about how that was done (see comments on PDF). Something that would be complementary and quite standard in 3D GM analyses when a phylogenetic backbone is available is to include the phylogeny in morphospace (just for the amphisbaenians, rather than the larger squamate sample you have). This assists with visualizing the shape differences across taxa, and more easily highlights shapes and features that are convergent across taxa, at a glance. It's a little different than mapping discrete characters onto phylogeny, and instead is a way to explore the evolution of ear shape overall, phylogenetically, something that can't be captured in a matrix. You can also test for phylogenetic correspondence of shape (i.e., ear features due to ancestry rather than ecology) within amphisbaenians using Geomorph, once you have that phylogeny incorporated into R. This would be an interesting aspect to include because convergence across amphisbaenians is mentioned a few times in the paper, but not really explored. It may help to bolster predictions made in the discussion regarding different substrate use, too.

- For the proportional size differences of the extracolumella (line 188): please show these data -- so far only the skull lengths are reported, not the proportional differences, nor the extracolumella lengths. Adding columns to the table of specimen data would suffice, or this can be part of supp table 2. Also, for the discussion later, is this trend potentially ontogenetic variation, or do you suspect all of your specimens to be adults? It would probably be a 'best practice' to talk about the state of your specimens in the methods, too. I know it's tough for understudied taxa, but you can say something like "to the best of our knowledge all specimens were adults," OR you can lean into the unknown and state clearly that neither ontogentic age nor sex was known for most specimens due lack of these data upon field collection (or because baenies have been severely understudied, we don't know anything about pop level variation). The trend is an interesting observation, so the more you can say (even to acknowledge limitations!), the more useful it will be.

- later around line 252 consider reporting measurements for the other proportions discussed.

You could also use the centroid sizes and do a statistical regression across all taxa to test for the effects of size/ontogeny, to see if it is a trend across all taxa you studied and not only within Zygaspis. [Ah ok, this was reported farther down at line 293 -- I amend my suggestion here to recommend that you mention this test in the methods]

- the columella/extracolumella descriptions are somewhat disorganized in terms of when anatomy is introduced vs. when it is explained (i.e., the sequencing of descriptions). I made suggestions in the annotated PDF and recommend that you read through all the comments in that section before editing. Some items I thought were missing turned out to be present farther down in the MS, so the earlier comments may be less relevant, or an alternate revision may be needed to improve the flow of the text.

- for Figure 1, this is not at all a requirement, but a collegial suggestion if it's not too big an obstacle: change the isolated columella views to lateral/horizontal. Without reading the legend carefully, at first I assumed they would be shown in situ as with the braincase horizontal (and thus the lateral surface of the bone). This is the most accessible view to most researchers, so it is the one the brain expects to see. Even for someone who looks at CT data a lot, I had to remind my brain the isolated views were of the ventral surface, and then rotate them in my head to decide which way the shafts were going. Or show both views -- stapes are underappreciated anyway!

- along with the above, I think more description needs to be added to the '4 categories' of columella orientation in the text. They all kind of have a medial aspect to them though, don't they? I would clarify this a bit based on R. floridana in Fig 1. In that specimen, footplate and the shaft (though not as much as in Z. quad) are deflected anterolaterally. In other words they are not located strictly perpendicular to the braincase. It's vague to say the footplate 'faces medially' because yes, the wide flat surface faces medially in taxa with a footplate parallel to the skull, but here in Rhinuera it has some posterior deflection. Also it's usually the more lateral surface that is the leading descriptor in papers because it's the side most visible to researchers; you may need to revise the way you describe the footplate and use terms more like perpendicular, parallel, angled (i.e., more points of reference are needed). The other important point here is that the Rhineura columella is not the most 'basic' or least deflected. It's actually quite angled from horizontal. So this may mean adjusting the other 3 category descriptions, too.

- based on figure 4, I would qualify these all as compact (as stated in the introduction generally for baenies), considering inner ear to vestibule ratio. What does stand out is the slenderness of the canals in many amphisbaenians. But for potentially "non-compact" (or less compact) baenie inner ears, please compare to the python depicted by Gray 1908 and the many non-burrowing squamates shown in Yi 2015; those are much less compact than any baenies depicted. In non fossorial snakes the vestibule is relatively small compared to the three semicircular canals together, whereas in your baenies, the canals closely track the space taken up by the vestibule even if the structures don't make contact. In fact, uropeltid snakes have 'compact' semicircular canals that also retain a small gap with the vestibule (Olori, 2010 fig 6). In many ways, the baenie ears looks more like a uropeltid, than a uropeltid resembles a non-fossorial snake. To be fair, I am most familiar with snakes, so I am not exactly sure what an 'average' squamate vestibule to canal ratio looks like. Still, the idea that amphisbaenians have "compact" ears is backed up by their position on your PC1 axis, which implies a rounded shape with a larger vestibule. PC1 suggests that amphisbaenians might be the 'most' compact, with a highly spherical vestibule and highly rounded canals, among the squamates you surveyed.

- line 223 - very cool observation about the inclination of the lateral canal matching that of the snout in Rhineura! Does not occur in any other taxa? This actually seems really cool and would match predictions made for mammals regarding head orientation and semicircular canal shape! Consider testing/exploring this quantitatively? It would be applicable for testing ecological predictions of extinct taxa in future if you could establish a trend.

- line 227 - thinness of canals: True there is a lot of variation here and they don't approach the extreme slenderness of say Dinylesia. But again, overall, they are moderately slender relative to the size of the vestibule. Did you compare these across baenie taxa quantitatively? If so, please show those data and the proportions, and provide more context from prior work on other clades. In other words, describe how 'slenderness' was assessed in your study so that future work can include your data in a comparative context.

- line 235: but what about panel B, Bipedidae? Or perhaps I'm looking at the wrong orientation. Maybe add a little description here, such as from which view it looks most right-angled. Ant canal does dip below horizontal in Bipes, so maybe that disqualifies it from being 90 deg.

-line 281 to 286: PCs of ear shape: see comment on PDF about adjusting description of positive PC1 shape. Also note in your PC2 image that the heighest height of the post and ant canals has also increased for positive PC2 vs. negative PC2. The distance between the vestibule and the canal has increased dorsally, so the canals are taller (lengthened vertically) in both directions (up and down), not simply extended down. This is cool, it nicely shows exteme snake shapes that are most drastic in tree dwelling taxa, aligning strongly with your "arboreal" taxa in the figure! (again take a look at the Gray 1908 python ear). This is a really neat computational confirmation of that "gut feeling" that arboreal taxa (snakes anyway) also stand out.

- line 302 and rest of the character list: please thoroughly check and update the figure and panel references. In many cases they seem to be reversed from the example that is intended (i.e., state 0 matches the figure reference for state 1, and vice versa).

- line 337: I've only seen Liotyphlops so I am unfamiliar with ear variation across the blindsnakes; however that genus has extremely thick canals and some of 'scolecophidian' disparate morphology could be more related to their very tiny skull size rather than ecology. Moreover, the blindsnakes feed differently than other snakes (and have seriously alien skulls as a result). One thing to look into is whether blindsnakes are actively digging through hard substrates (like uropeltids and baenies); check Cundall and Irish 1993, a paper on Anomochilus if I recall, may talk about that. They are fossorial in ecology for sure, but are they also actively head first burrowers in terms of locomotion and biomechanics, or do they use existing tunnels and/or move through soft detritus (I know that some even climb trees!)? This could lead to different stresses on the skull but really we have no idea, it hasn't been tested yet! Still, before using the blindsnakes to frame your baenie results, it's worth thoroughly checking their locomotion style and updating the discussion as needed [in fact, either answer would make your discussion more interesting, just in different ways]. Also, the molecular contingent might take insult at the use of Scolecophidia without quotes; most recent analyses agree they are paraphyletic to some degree.

- line 352: Gans suggested that relatively larger otic capsules was related to size differences; in very small taxa the ears still have to maintain a certain functional size. In fossorial taxa in particular, this is not related to hearing but rather balance. The sacculus often contains a mass of calcite crystals (statolithic mass) that might be related to sensing gravity (up/down) when moving in 3D through tunnels; it's often much larger in fossorial taxa and perhaps it's the statolith driving the size of the vestibule to some degree. In any case, I would be cautious in speculating here that 'less restrictive' movement might be related to expanded otic capsules, without testing. It's an interesting idea, but this sentence in isolation covers up alternative hypotheses that have been discussed more in prior work (but that also haven't been fully tested).

- line 370: caecilian comparisons: this conflicts with your statement in the sentence before, regarding shared similarity. In fact more similarities are listed than differences. But also there is probably more opportunity here to expand this comparison and help frame your results. Breaking free of squamate shared ancestry actually may be more enlightening for discussing convergence than making comparisons to only other squamate clades.

- I strongly recommend being more careful (precise?) throughout the paper in applying fossorial vs. burrowing. They are often used interchangeably, but actually don't always overlap in meaning. For example, taxa can live underground without making their own burrows. Or they can be capable of digging, but don't really live underground. The key is being clear from the outset (Introduction) in how you will use the terms, and adding more description about the habitats (location) vs. habits (locomotion) of the clades you discuss instead of relying on single terms.

-overall the Discussion seems short. It is rightly focused on ecology and potential applications for biomechanics, fossil interpretation, and questions related to squamate origins and phylogeny. However, there are some really great opportunties here for more broadly addressing variation! You have amassed an impressive data set of inner ears within a bizarre and important clade and more should be written about the variation across and within baenie taxa specifically. What else can be said about individul variation? Potential ontogenetic variation? You also did quite a bit of phylogenetic work, which somewhat 'speaks for itself' but could be put into wider or more applied context in the Discussion. For example do any of the proposed characters support or contradict conflicting hypotheses of amphisbaenian phylogeny? Do the ear characters reveal, or confirm, or reject evolutionary trends proposed within Amphisbaenia?

Note: I did not check the references.

Thank you very much for sharing your work and I look forward to seeing the finished product. We need more baenie studies. Best wishes, Jen Olori

6. PLOS authors have the option to publish the peer review history of their article (what does this mean?). If published, this will include your full peer review and any attached files.

Reviewer #1: No

---

## [Author Response · Author response to Decision Letter 0]

17 Aug 2024

Dear Dr. Ulrich Joger

Heere is my rebuttal to the comments, all my responses are in bold letters. There is also a response to many more comments on the pdf. 

- Line 2: Short title: The inner ear. 

Changed

- Line 110: in a Zeiss Xradia micro-CT (too many capital letters). 

Changed.

- Line 179: develop in the genus 

Changed.

- Line 329/330: The detected diversity should not only have utility in phylogenetic analysis, but should also be interpreted in a functional context. Please try to start such an interpretation in the light of different burrowing techniques of the studied genera. The discussion deserves an enlargement.

We are limited by the knowledge on these amphisbaenians, but we correlted this with well known head shape and digging style. 

- Line 336: workers have also demonstrated. 

Changed.

- Line 342: of the 

Redundant words removed

- Line 347: with some species 

Word chaged, thanks for spotting this

- Line 356: describe accurately 

Word order swapped.

- Line 373: diversity of 

changed

- Line 375: "might help to improve our knowledge on fossorial ecologies": This sounds too general. Please try to give some hypotheses how the inner ear is connected to specific ecologies (see above). Convergencies with fossorial snakes should also better highlighted.

This discussion is now expanded and better comparison with fossorial snakes was made.

References: Refs 16 and 25 are identical, also refs 13 and 17. Please change numbering.

Figure Captions: Line 541: Rhineura floridana; 

Changed

 - Line 543: position of the right inner ear

Changed 

We look forward to receiving your revised manuscript.

Kind regards,

Ulrich Joger

Academic Editor

PLOS ONE

Journal Requirements:

"The Carl Gans Fund, Wilson-Warner, James D. Long and the Biology Department Scholarships supported Geneva E. Clark. Sam Houston State University Enhancement Research Grant (2014) to Juan D. Daza, Patrick J. Lewis, Monte L. Thies supported obtaining data for this study and funds from the Jackson School of Geosciences at The University of Texas at Austin to Christopher J. Bell."

"We thank Jessie Maisano and Matt Colbert from the University of Texas – High Resolution X-Ray CT Facility for scanning most of the specimens used in this study. This project started as an independent undergraduate research project from GEC in 2017 at Sam Houston State University, and gradually growth into a larger survey. Funding was received from multiple agencies, but we want to specially thank The Carl Gans Fund for sponsoring GEC to present part of this work in the 2016 meeting of the American Society of Ichthyology & Herpetology."

"The Carl Gans Fund, Wilson-Warner, James D. Long and the Biology Department Scholarships supported Geneva E. Clark. Sam Houston State University Enhancement Research Grant (2014) to Juan D. Daza, Patrick J. Lewis, Monte L. Thies supported obtaining data for this study and funds from the Jackson School of Geosciences at The University of Texas at Austin to Christopher J. Bell."

Additional Editor Comments:

The reviewer suggested a number of valuable improvements for your manuscript. Please try to follow them, especially the discussion should be enlarged as she proposed.

Reviewers' comments:

Reviewer's Responses to Questions

Comments to the Author

1. Is the manuscript technically sound, and do the data support the conclusions?

Reviewer #1: Yes

2. Has the statistical analysis been performed appropriately and rigorously? 

Reviewer #1: Yes

3. Have the authors made all data underlying the findings in their manuscript fully available?

Reviewer #1: No

4. Is the manuscript presented in an intelligible fashion and written in standard English?

Reviewer #1: Yes

5. Review Comments to the Author

Reviewer #1: This study represents a really neat example of using CT technology to access anatomical systems that are not usually observable, in taxa that also are not often easily observable! I also appreciate that qualitative comparisons were not overlooked, but utilized in tandem with popular 3D geometric morphometric methods.

Overall the paper was well-written with clear anatomical descriptions and analysis. The syntax and grammar can be improved in many places, and all wording suggestions, typo edits, requests for citations, etc. have been annotated directly onto the attached PDF, along with all comments and questions. There are also areas of text where some re-structuring or re-organization will be necessary to improve the flow and clarity of the paper. The full resolution CT and other imagery was sharp and clear, a real strength of the paper; the color coding is clever and helpful for following along in the figures.

The more major revisions are repeated and highlighted here for emphasis:

-abstract: suggest including more of the specific (and useful!) results observed in this study.

More information from the study added. 

- line 67: innervation and shaft morphology not really discussed in the Results though mentioned here as needing study; can more info about the shafts and foramina be added to the paper? The columellas are nicely figured.

one of the coathors is doing a chapter of his thesis on the columella of Amphisbaenians, so we prefer to leave this outside. The main topic of the paper if the inner ear, and we included just some brief reference to this structure to give the reader more context. 

- The mapping of new characters onto the amphisbaenian tree is interesting and could be useful in future. Please add a little more detail about how that was done (see comments on PDF). Something that would be complementary and quite standard in 3D GM analyses when a phylogenetic backbone is available is to include the phylogeny in morphospace (just for the amphisbaenians, rather than the larger squamate sample you have). This assists with visualizing the shape differences across taxa, and more easily highlights shapes and features that are convergent across taxa, at a glance. It's a little different than mapping discrete characters onto phylogeny, and instead is a way to explore the evolution of ear shape overall, phylogenetically, something that can't be captured in a matrix. You can also test for phylogenetic correspondence of shape (i.e., ear features due to ancestry rather than ecology) within amphisbaenians using Geomorph, once you have that phylogeny incorporated into R. This would be an interesting aspect to include because convergence across amphisbaenians is mentioned a few times in the paper, but not really explored. It may help to bolster predictions made in the discussion regarding different substrate use, too.

We added more details about the mappin, we implemented ACCTRAN and DELTRAN and exloed the phylogenetic signal on the morphospace.

- For the proportional size differences of the extracolumella (line 188): please show these data -- so far only the skull lengths are reported, not the proportional differences, nor the extracolumella lengths. Adding columns to the table of specimen data would suffice, or this can be part of supp table 2. Also, for the discussion later, is this trend potentially ontogenetic variation, or do you suspect all of your specimens to be adults? It would probably be a 'best practice' to talk about the state of your specimens in the methods, too. I know it's tough for understudied taxa, but you can say something like "to the best of our knowledge all specimens were adults," OR you can lean into the unknown and state clearly that neither ontogentic age nor sex was known for most specimens due lack of these data upon field collection (or because baenies have been severely understudied, we don't know anything about pop level variation). The trend is an interesting observation, so the more you can say (even to acknowledge limitations!), the more useful it will be.

Measurements in the genus Zygaspis were added.

- later around line 252 consider reporting measurements for the other proportions discussed.

You could also use the centroid sizes and do a statistical regression across all taxa to test for the effects of size/ontogeny, to see if it is a trend across all taxa you studied and not only within Zygaspis. [Ah ok, this was reported farther down at line 293 -- I amend my suggestion here to recommend that you mention this test in the methods]

Done

- the columella/extracolumella descriptions are somewhat disorganized in terms of when anatomy is introduced vs. when it is explained (i.e., the sequencing of descriptions). I made suggestions in the annotated PDF and recommend that you read through all the comments in that section before editing. Some items I thought were missing turned out to be present farther down in the MS, so the earlier comments may be less relevant, or an alternate revision may be needed to improve the flow of the text.

Section re-organized

- for Figure 1, this is not at all a requirement, but a collegial suggestion if it's not too big an obstacle: change the isolated columella views to lateral/horizontal. Without reading the legend carefully, at first I assumed they would be shown in situ as with the braincase horizontal (and thus the lateral surface of the bone). This is the most accessible view to most researchers, so it is the one the brain expects to see. Even for someone who looks at CT data a lot, I had to remind my brain the isolated views were of the ventral surface, and then rotate them in my head to decide which way the shafts were going. Or show both views -- stapes are underappreciated anyway!

We prefer the way the figure is now. 

- along with the above, I think more description needs to be added to the '4 categories' of columella orientation in the text. They all kind of have a medial aspect to them though, don't they? I would clarify this a bit based on R. floridana in Fig 1. In that specimen, footplate and the shaft (though not as much as in Z. quad) are deflected anterolaterally. In other words they are not located strictly perpendicular to the braincase. It's vague to say the foo

---

## [Editor Report · Decision Letter 1]

16 Sep 2024

PONE-D-24-09623R1The unique inner ear labyrinth of worm-lizards (Amphisbaenia: Squamata)PLOS ONE

Dear Dr. Daza,

Thank you for submitting your manuscript to PLOS ONE. After careful consideration, we feel that it has merit but does not fully meet PLOS ONE’s publication criteria as it currently stands. Therefore, we invite you to submit a revised version of the manuscript that addresses the points raised during the review process.

In fact, your manuscript has been substantially improved, and the reviewer's critique has been followed in a satisfactory degree. I only found a few typographical errors (see below). Yet please look at the complete text carefully, as there may be more errors hidden in the many revised passages.

We look forward to receiving your revised manuscript.

Kind regards,

Ulrich Joger

Academic Editor

PLOS ONE

Journal Requirements:

Additional Editor Comments:

Here are a few minor corrections which should be done (lines numbered according to the revised manuscript):

- L. 38 - reptiles

- L. 217 - delete 'in a' (double)

- L. 260 - delete 'skull length'

- L. 407 - potentially

- L. 471 - delete 'useful' (double)

- L. 505 - replace 'not possible decribe' by the previous wording 'difficult to describe'

---

## [Author Response · Author response to Decision Letter 1]

23 Sep 2024

Dear Dr. Ulrich Joger

Here is my rebuttal to the comments, all my responses are in bold letters. There is also a response to many more comments on the pdf. We remove all the changes indiced by you:

- L. 38 - reptiles

- L. 217 - delete 'in a' (double)

- L. 260 - delete 'skull length'

- L. 407 - potentially

- L. 471 - delete 'useful' (double)

- L. 505 - replace 'not possible decribe' by the previous wording 'difficult to describe'

We also correted a few more typos, indicated in the new marked copy.

---

## [Editor Report · Decision Letter 2]

2 Oct 2024

The specialized inner ear labyrinth of worm-lizards (Amphisbaenia: Squamata)

PONE-D-24-09623R2

Dear Dr. Daza,

We’re pleased to inform you that your manuscript has been judged scientifically suitable for publication and will be formally accepted for publication once it meets all outstanding technical requirements.

Kind regards,

Ulrich Joger

Academic Editor

PLOS ONE
---

## [Editor Report · Acceptance letter]

7 Oct 2024

PONE-D-24-09623R2 

PLOS ONE

Dear Dr. Daza, 

I'm pleased to inform you that your manuscript has been deemed suitable for publication in PLOS ONE. Congratulations! Your manuscript is now being handed over to our production team.

Kind regards, 

on behalf of

Dr. Ulrich Joger 

Academic Editor

PLOS ONE